# Outcomes of children with clear cell sarcoma of kidney following NWTS strategies in Shanghai China (2003–2021)

**Anan Zhang**[1], **Xiaojun Yuan**[2], **Shayi Jiang**[3], **Dongqing Xu**[2], **Can Huang**[3], **Jing yan Tang**[1], **Yijin Gao**[1]*

**1** Department of Hematology and Oncology, Shanghai Children's Medical Center, Shanghai Jiao Tong University School of Medicine, Shanghai, China, **2** Department of Pediatric Hematology and Oncology, Xinhua Hospital Affiliated to Shanghai Jiao Tong University School of Medicine, Shanghai, China, **3** Department of Hematology and Oncology, Children's Hospital of Shanghai, Shanghai Jiao Tong University School of Medicine, Shanghai, China

* gaoyijin@scmc.com.cn

**Data Availability Statement:** All relevant data are within the manuscript and its Supporting information files.

## Abstract

### Background

Although clear cell sarcoma of kidney (CCSK) is rare, it is the second most common renal tumor in children after Wilms' tumor. NWTS and SIOP are two major groups which had made tremendous efforts on renal tumors, but the strategies are different, for NWTS follows the upfront surgery principle providing definite pathology and the SIOP follows the upfront chemotherapy principle, each has its own advantages. Here we aimed to evaluate the outcomes of CCSK in China following NWTS strategies to analyze the prognostic factors.

### Methods

For this multicenter retrospective study, a total of 54 patients were enrolled from three children's hospitals, between April 2003 and December 2021. Treatment comprised upfront radical nephrectomy, followed by radiotherapy and intensive chemotherapy. Clinical records were regularly updated. Prognostic factors and survival rates were evaluated.

### Results

The 54 enrolled patients had a median age of 37 months (range, 4 months to 11.4 years). The stage distribution was 16% stage I (n = 9), 30% stage II (n = 16), 39% stage III (n = 21), and 15% stage IV (n = 8). Among stage IV, metastasis sites included the lung (n = 6), bone (n = 1), and intra-orbital/cervical lymph node (n = 1). After a median follow-up of 5.6 years, the 5-year event-free survival (EFS) was 82.4±5.4%, and overall survival was 88.1±4.6%. The EFS was 100% for stage I, 93.8 ±6.1% for stage II, 71.1±10.0% for stage III, and 68.6 ±18.6% for stage IV. Univariate analysis revealed that staging (III/IV), tumor rupture, and inferior vena cava tumor thrombus were inferior prognostic factors. Multivariate analysis revealed that tumor rupture was independent poor prognostic factor ($P$ = 0.01, $HR$ 5.9). Among relapsed patients, relapse occurred a median of 11 months after diagnosis (range,

**Funding:** The author(s) received no specific funding for this work.

**Competing interests:** No authors have competing interests.

4–41 months), and 50% (4/8) achieved a second complete remission after multiple treatment. None of the six lung metastasis patients received lung RT, only one patient developed a relapse and was salvaged by RT after relapse.

## Conclusions

Tumor rupture was independent poor prognostic factor. Upfront surgery of NWTS strategies can make a definite pathology diagnosis, but how to reduce tumor rupture during surgery is important especially in developing countries. The outcomes of patients with stage I–III CCSK in China were comparable to findings in other developed countries. Better outcomes were achieved in stage IV CCSK by using an intensive chemotherapy regimen including carboplatin, which require further confirmation by AREN0321. Lung RT may be safely omitted in selected patients who achieve a compete radiographic response after 6 weeks of systemic treatment (including surgery). Treatment should be encouraged even in CCSK cases with metastasis and relapse.

## Introduction

Clear cell sarcoma of kidney (CCSK) is a rare tumor, accounting for 5% of primary renal tumors in children, but it is the second most common renal malignancy, after Wilms' tumor (WT) [1–3]. Historically, CCSK was considered an unfavorable histologic type of WT, with poor survival. In 1970, Kidd recognized it as a separate clinicopathologic renal tumor, with aggressive behavior and a propensity for bone metastases, which led it to be named "bone metastasizing renal tumor of childhood". Beckwith and Palmer later designated it as "clear cell sarcoma of kidney" [4, 5]. CCSK most often presents at 2–4 years of age, and shows male predominance. About 4–12% of patients present with metastatic disease at diagnosis [2, 6, 7].

The National Wilms Tumor Study (NWTS) and The International Society of Pediatric Oncology (SIOP) Nephroblastoma Trial Group are two major groups which had made tremendous efforts on renal tumors, but the strategies are different, each has its own advantages. In NWTS, standard treatment of CCSK consists of upfront radical nephrectomy with definite pathology diagnosis, followed by radiotherapy and chemotherapy. Historically, CCSK had inferior survival compared to WT; however, outcomes have dramatically improved in recent years with the use of more intensive chemotherapy regimens. Notably, in the NWTS-3 trial, the additional use of doxorubicin improved 6-year event-free survival from 25% to 64% [2, 8]. In the NWTS-5 study, patients were administered four-agent chemotherapy called regimen I (including cyclophosphamide, etoposide, vincristine, and doxorubicin) for 6 months, along with radiotherapy, regardless of disease stage [1, 9]. The five-year relapse-free survival rate was 79%, and overall survival (OS) was 89% [10]. The SIOP Renal Tumor Study differed from NWTS in terms of advocating for pre-operative chemotherapy to facilitate surgery [2, 11]. Specifically, SIOP advocates for preoperative chemotherapy in all children, aged 6 months to 16 years, with a radiographically consistent Wilms tumor diagnosis. Since this protocol involves a degree of misdiagnosis, the current SIOP consortium also performs biopsy or fine needle aspiration in cases of imaging or clinical atypia [12]. In 2013, SIOP reported their largest study, involving 191 CCSK patients treated with SIOP 93–01. Pre-operative chemotherapy was administered to 169/191 patients. The five-year EFS and OS rates for CCSK were 79% and 86%, respectively [12].

Studies of CCSK are limited by the disease rarity, with scarce data available from groups other than NWTS and SIOP. Since 2003, the Renal Cancer Research Group of the Chinese Children's Cancer Group (CCCG) has regularly issued pediatric renal tumor protocols, all of which are modified from NWTS and combined with Chinese conditions, including the CCCG-WT-2003, WT-2009, WT-2015, and WT-2019 protocols. Although reports have described the outcomes of CCCG Wilms' tumors, the patients of CCCK have never been summarized.

In the present multicenter retrospective study, CCSK was investigated at three Shanghai Children's hospitals (Shanghai Children's Medical Center, Shanghai Xinhua Hospital, and Children's Hospital of Shanghai), all of which followed the CCCG-WT protocols. Here we aimed to evaluate the characteristics and outcomes of CCSK treated following NWTS strategies in China, with the goal of further improving the outcomes of CCSK.

## Materials and methods

### Patient population

Children of ≤16 years of age, with newly pathology-confirmed clear cell sarcoma of kidney were eligible for trial CCCG-WT. From April 2003 to December 2021, 62 patients were registered at three children's hospitals: Shanghai Children's Medical Center, Shanghai Xinhua Hospital, and Children's Hospital of Shanghai. Eight patients were excluded based on the following criteria: erroneous diagnosis (n = 2), previous treatment (n = 2), refusal of further protocol therapy by parents (n = 4). The flow chart of patient has been presented. (Fig 1) The study protocol was approved by the Ethical Review Board of Shanghai Children's Medical Center (SCMCIRB-J2016002), Shanghai Xinhua Hospital, and Children's Hospital of Shanghai, and written informed consent was obtained prior to enrollment from all parents. All the participants have been written informed that the data will be used for analysis and collection. Clinical records were regularly updated since the beginning of study and assessed the data for collection in 2022.6.1. The study is also registered with ChiCTR-OPC-15006533.

### Diagnosis and staging

Immediate nephrectomy was performed to ensure accurate histologic diagnosis, as advocated by NWTS. If the tumor could not be completely resected, a tumor biopsy was taken to obtain pathology. The NWTS Staging system was used [13, 14]. At the time of diagnosis, patients underwent a staging work-up, consisting of physical examination; abdomen computed tomography (CT); and/or magnetic resonance imaging (MRI), chest CT, brain MR, bone marrow aspirate, ultrasonography, bone scan, or PET-CT.

### Stratification and treatment

The treatment protocol comprised surgery, chemotherapy, and radiotherapy. All patients received postoperative radiation therapy (RT) to the primary tumor bed. Radiation therapy was started within 10 days from surgery. RT was only considered necessary in cases with lung metastasis residue after 6 weeks. For those patients with complete radiographic response of the lung metastasis after 6 weeks of systemic therapy including surgery did not receive lung RT. Patients younger than 6 months of age did not receive radiotherapy.

There were two risk groups. Patients with stage I–III disease were considered the standard-risk group, and received the WTSG-5-I protocol, which included doxorubicin, vincristine, cyclophosphamide, and etoposide for 6 months [9, 14]. Patients with metastatic CCSK were

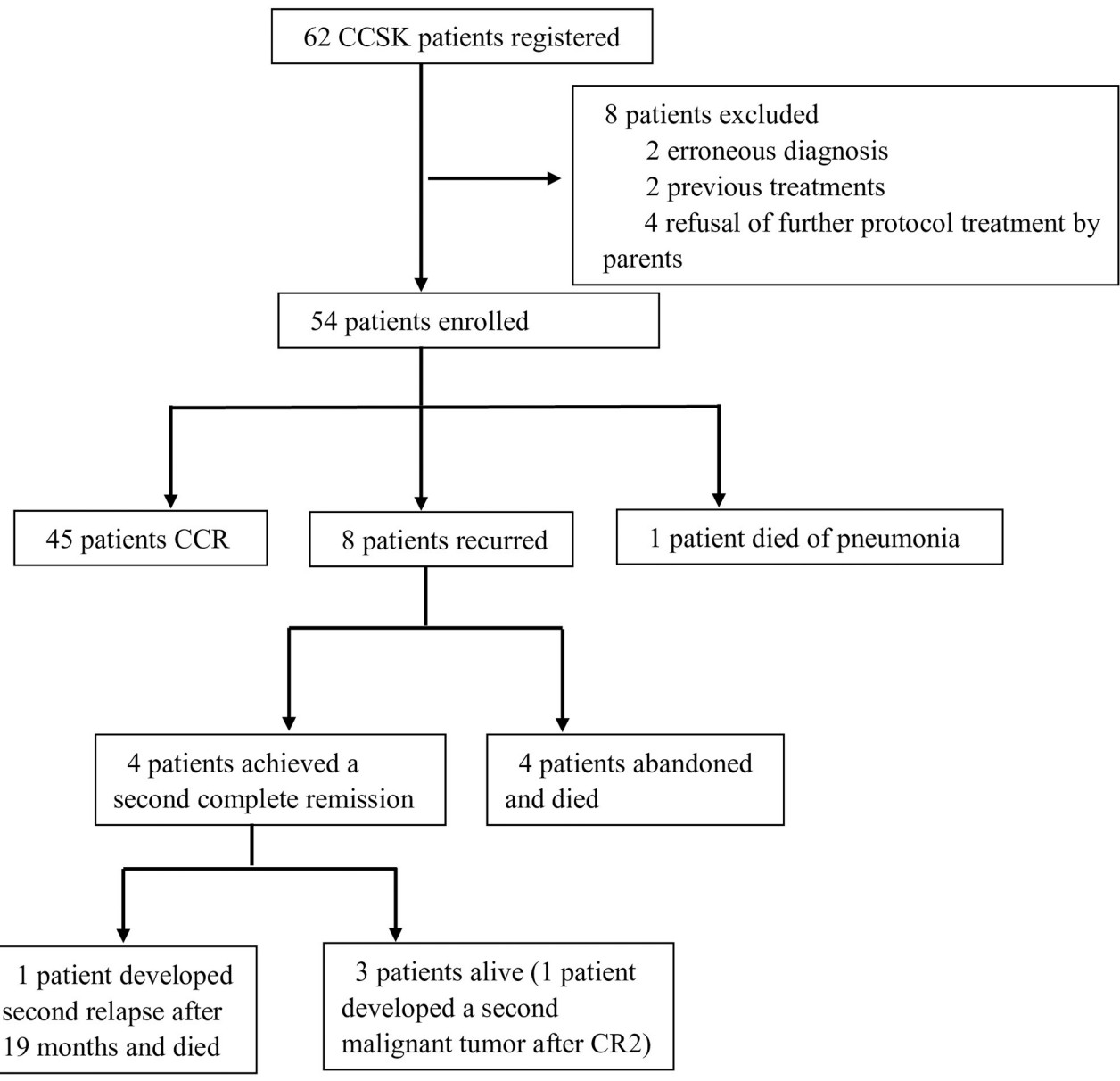

CR: complete remission

CCR: continuous complete remission

**Fig 1. Flow chart.**

the high-risk group, and received chemotherapy including carboplatin, etoposide, cyclophosphamide, doxorubicin, and vincristine (with variable dosages). Since 2019, we have followed the AREN0321 study, which used a modified UHI regimen aiming to improve the outcomes of metastatic CCSK and reduce the non-hematologic toxicities [1, 15]. The chemotherapy regimen is presented in Table 1.

**Table 1. Chemotherapy regimens of clear cell sarcoma.**

| Staging | Regimen | Drugs | Patient Numbers |
|---|---|---|---|
| I-III | WTSG-5-I (13, 14) | Doxorubicin, vincristine, cyclophosphamide, etoposide | 46 |
| IV | PE/CDV (13) | Carboplatin, etoposide, cyclophosphamide, doxorubicin, vincristine | 6 |
| | Revised UH-1 (15) | Carboplatin, etoposide, cyclophosphamide, doxorubicin, vincristine | 1 |
| | WTSG-5-I | Doxorubicin, vincristine, cyclophosphamide, etoposide | 1 |
| Chemotherapy before nephrectomy (before 2019) | IEV | Ifosfamide, etoposide, vincristine | 4 |

The details of regimen can see references (13, 14, 15, 16).

## Statistical method

Event-free survival (EFS) was measured from the day of diagnosis to an event (progression, relapse, second malignancy, or death from any cause). The last follow-up was in June 2023. Patients lost to follow-up after treatment were censored at the time of their last follow-up. Kaplan–Meier plots were generated for EFS and overall survival. Associations between EFS or OS and other factors were evaluated using a log-rank test for univariate analysis and Cox regression for multivariate analysis. Statistical analyses were performed using R software.

# Results

## Patient characteristics

A total of 54 CCSK patients were enrolled from three children's hospitals, between April 2003 and December 2021. Among these patients, 37 were male and 17 were female, and their ages ranged from 4 months to 11.4 years (median, 37 months) at the time of diagnosis. Diagnostic work-ups revealed 28 patients with right-sided tumors, and 26 with left-sided tumors. According to the NWTS staging system, 9 patients were stage I, 16 stage II, 21 stage III, and 8 stage IV. Among stage IV, metastasis sites included lung (n = 6), bone (n = 1), and intra-orbital/cervical lymph node (n = 1).

## Treatment

Nephrectomy at diagnosis was performed in 46 patients, of whom 35 underwent lymph node dissection (30/35 were node-negative and 5/35 were node-positive). Renal tumor biopsy was performed in 7 patients. One patient was seriously ill with tumor rupture and poor general condition; this patient did not undergo biopsy, but received surgical resection after two cycles of chemotherapy, and the pathology confirmed CCSK. During surgery, the maximum tumor size was 17.3×11.6×15.6 cm. The margin status was complete in 29 cases, incomplete in 6 cases, and unknown in 11 cases. Inferior vena cava tumor thrombus was present in 6 cases, and peri-operative tumor rupture occurred in 5 cases, and four cases were intraoperative tumor spill. The median tumor size of **upfront resection** was 10cm (range 9.4–14.5), and the median tumor size of **tumor spill** was 13.1cm (11–18).

The chemotherapy regimen is summarized in Table 1. Overall, 50 patients received radiotherapy as planned, and 4 patients did not receive radiotherapy for personal reasons (2 of these patients were stage I). Radiotherapy was started a median of 12 days after surgery (range, 4 days to 272 days). The RT dose was 10.8 Gy/6 Fx in 32 cases. The remaining 18 cases received a RT dose of 11–20 Gy, with a maximum dose of 20 Gy/10 Fx). None of the six patients with lung metastasis received lung radiotherapy. In one case, the lung metastasis was surgically

removed after chemotherapy, and in the other five cases, the lung metastases disappeared within six weeks after chemotherapy.

## Events

In the entire patient group (n = 54), the five-year EFS and OS were 82.4±5.4% and 88.1±4.6%, respectively, with a median follow-up of 5.6 years (range, 8 months–20 years) (Figs 2 and 3). Nine events occurred during follow-up: one patient died of pneumonia after chemotherapy, the other eight events were related to disease recurrence (Table 2). Relapse occurred a median of 11 months after diagnosis (4,41). Four patients abandoned further treatment after relapse and died. The other four patients received further treatment after relapse—including resection, intensive chemotherapy including carboplatin, and radiotherapy—and each achieved a second complete remission. After achieving a second CR after treatment, one patient developed a second relapse after 19 months (Table 2), and another patient (case 8) developed a second malignancy (Table 2). For six lung metastasis patients who did not receive lung RT, only one patient developed a relapse (CASE 3 in table) and was salvaged by RT after relapse and was now in second complete remission.

Up to the last follow-up in 2023, 10 patients had been lost to follow-up. Loss to follow-up occurred at a median of 63.5 months (range, 8 months to 106 months), with 6 patients being lost to follow-up after 60 months since diagnosis.

## Prognostic factors

The five-year EFS was 100.0% for stage I, 93.8±6.1% for stage II, 71.1±10.0% for stage III, and 68.6±18.6% for stage IV (Fig 4). The five-year OS was 100% for stage I, 93.8 ±6.1% for stage II, 78.6±9.8% for stage III, and 85.7±13.2% for stage IV. Univariate analysis revealed that inferior outcome was significantly correlated with stage III/IV disease ($\chi^2$ = 5.29, $P$ = 0.021), inferior

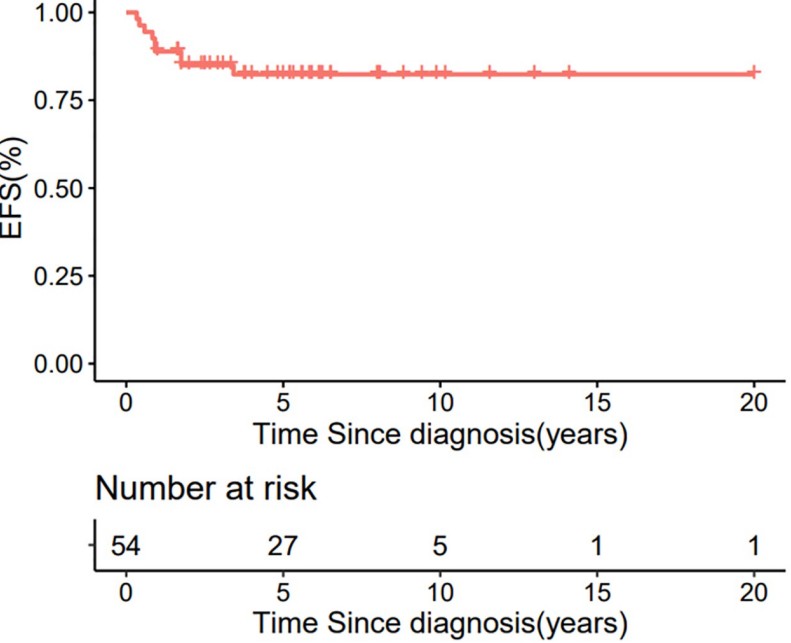

**Fig 2. Event free survival for all 54 CCSK patients.**

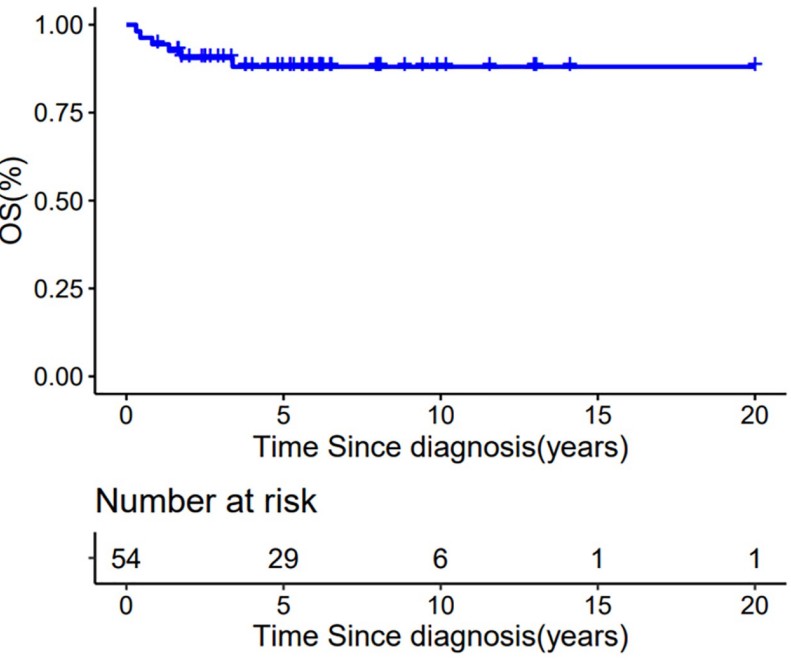

**Fig 3. Overall survival for all 54 CCSK patients.**

**Table 2. Characteristics and outcomes of patients with events.**

| Case Number | Age (month) | Staging | Event | Treatment after first progression/relapse | Outcome/Follow up time after relapse |
|---|---|---|---|---|---|
| CASE 1&2 | 9, 16 | III | Progression at 3.7 months, 10 months, respectively | Abandon after relapse | Death |
| Case 3 | 22 | IV, Lung metastases | Lung and ulna metastasis occurred at 41 months of follow-up. | Lung nodules resection, RT: 12Gy/8Fx, chemo of SIOP-UMBRELLA | DFS, 42 months in CR2 |
| Case 4 | 137 | III (severe intraoperative tumor rupture) | Diaphragm and liver metastasis after 21 months. | Surgery, RT, chemo of PE/CDV. 19 months later, metastasis of a bladder-rectal fossa mass. | Death after second relapse |
| Case 5 | 9 | III (intraoperative tumor rupture) | No RT at initial treatment. Abdomen recurrence at 6.8 months follow-up. | Receiving abdomen radiotherapy and chemotherapy of PE/CDV. | DFS, 156 months in CR2 |
| Case 6 | 47 | III (tumor rupture) | Inferior vena cava tumor thrombus residue after surgery. Lung metastasis occurred after 11 months. | Abandon after relapse | Death |
| Case 7 | 16 | II | Patient developed treatment-related pneumonia at 5 months and died. | Died | Death |
| Case 8 | 53 | III | Femoral metastases occurred at 10.5 months of follow-up. | Chemotherapy of UH1 regimen, no stem cell for transplantation. At 40 months of follow-up, the patient developed second malignancy (AML), and then receive chemotherapy +HSCT | DFS, 19 months after HSCT |
| Case 9 | 58 | IV, Intra-ocular metastasis | Recurrence at 21 months of follow ups | Abandon after relapse | Death |

HSCT: allogeneic hematopoietic stem cell transplantation

AML: Acute myeloid leukemia

RT: radiotherapy

CR2: a second complete remission after relapse or progression

DFS: disease free survival

PE/CDV: Carboplatin, etoposide, cyclophosphamide, doxorubicin, vincristine

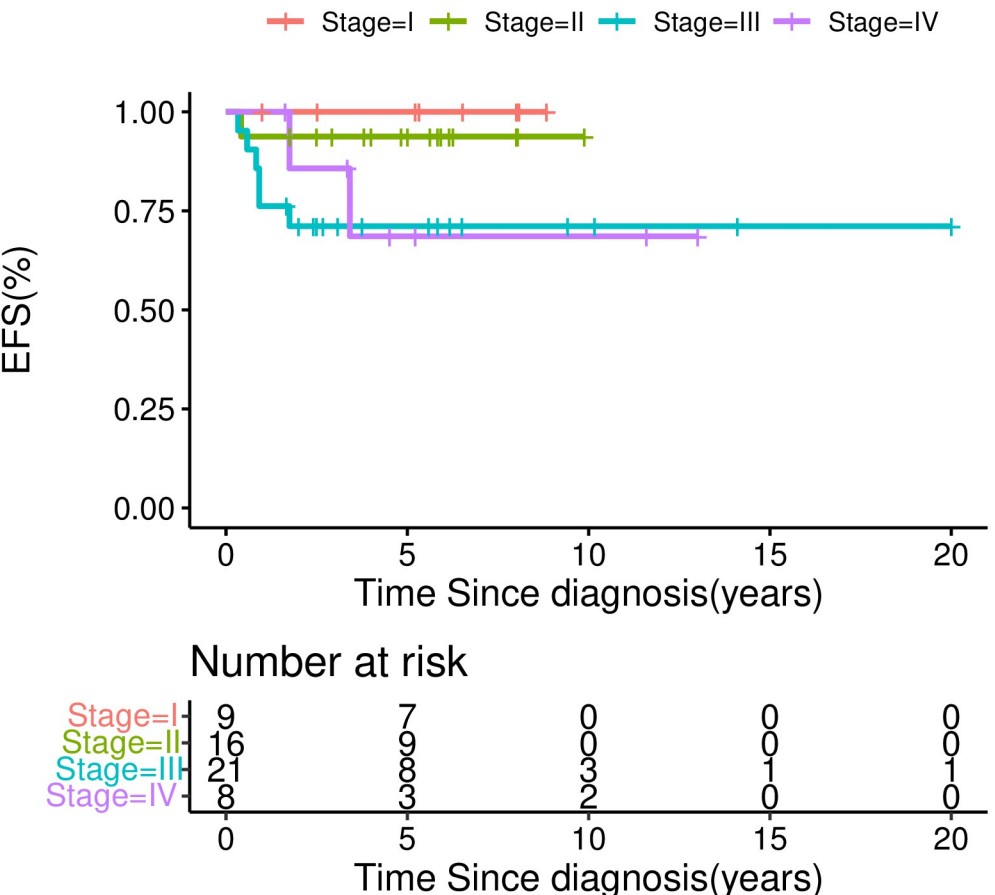

**Fig 4. EFS for CCSK patients by stage.**

vena cava tumor thrombus (41.7±22.2% *vs*. 87.4±4.8%, $\chi^2$ = 4.851, *P* = 0.028), and tumor rupture during or before surgery (40±21.9% *vs*. 86.8±5.1%, $\chi^2$ = 8.59, *P* = 0.003) (Figs 4–6). Multivariate analysis revealed that tumor rupture was independent poor prognostic factor (*P* = 0.01, *HR* 5.9, 95%Cl 1.4–24.5).

## Discussion

Intensive chemotherapy regimens have dramatically improved the outcomes of CCSK. For CCSK cases treated by NWTS and SIOP, the five-year EFS is 76–79%, and OS is 86–89% [10]. However, studies of CCSK are limited by the disease's rarity, such that there are scarce data available from groups other than NWTS and SIOP, especially from developing countries. Outside of NWTS and SIOP, the prognosis of CCSK has historically varied widely, with 5-year EFS rates ranging from 50–87.8% [7, 16–18].

Our present study was a retrospective multicenter investigation in China, and is the largest study in a developing country to date. It also included the third largest patient cohort published worldwide, with other reports often including only small patient groups [19]. Our results showed 5-year EFS and OS rates of 82.4±5.4% and 88.1±4.6%, respectively, which were significantly better than expected and comparable to those reported in NWTS and SIOP. Compared with the findings of NWTS-5, our results for stage III were slightly poorer (73% in

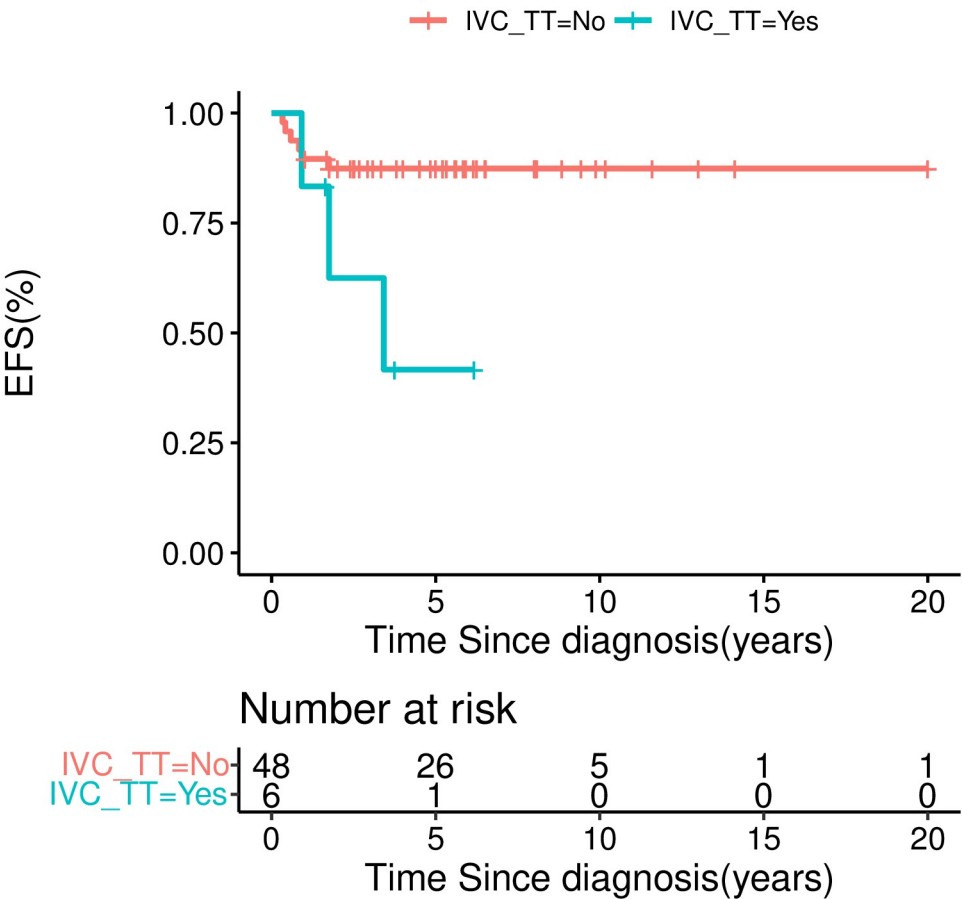

**Fig 5. EFS for CCSK patients by presence of inferior vena cava tumor thrombus or not.**

NWTS-5), but our results for stage IV were better (29% in NWTS-5), and were close to the result for stage IV in SIOP 93–01 (58.9%).

Our presently described good outcomes reveal the importance of following standard treatment in developing countries. Shanghai is rich in medical resources, making radiotherapy and timely diagnosis accessible. Two previous reports in the literature have demonstrated that lack of RT was a factor contributing to poor survival [20, 21]. Our findings are also consistent with a recent personal communication by the Shanghai CDC (Centers for Disease Prevention and Control) that described an 83% 5-year survival rate of childhood renal malignant tumors in Shanghai (2002–2016).

The stage III prognosis in our study may have been somewhat poor partly due to the higher proportion of tumor rupture (9.3%), Tumor rupture was independent poor prognostic factor. Upfront surgery of NWTS strategies can make a definite pathology diagnosis, but how to reduce tumor rupture during surgery is important especially in developing countries where surgeon is not experienced enough and there are often delays of diagnosis. It is known that disseminated tumor rupture can result in intraperitoneal tumor implantation, and it is recommended that operation be performed very carefully to avoid tumor spillage by experienced surgeons. Therefore, we think if the surgeons think the tumor has high risk of tumor spill, preoperative chemotherapy or needle biopsy maybe more preferred. In the future, we should

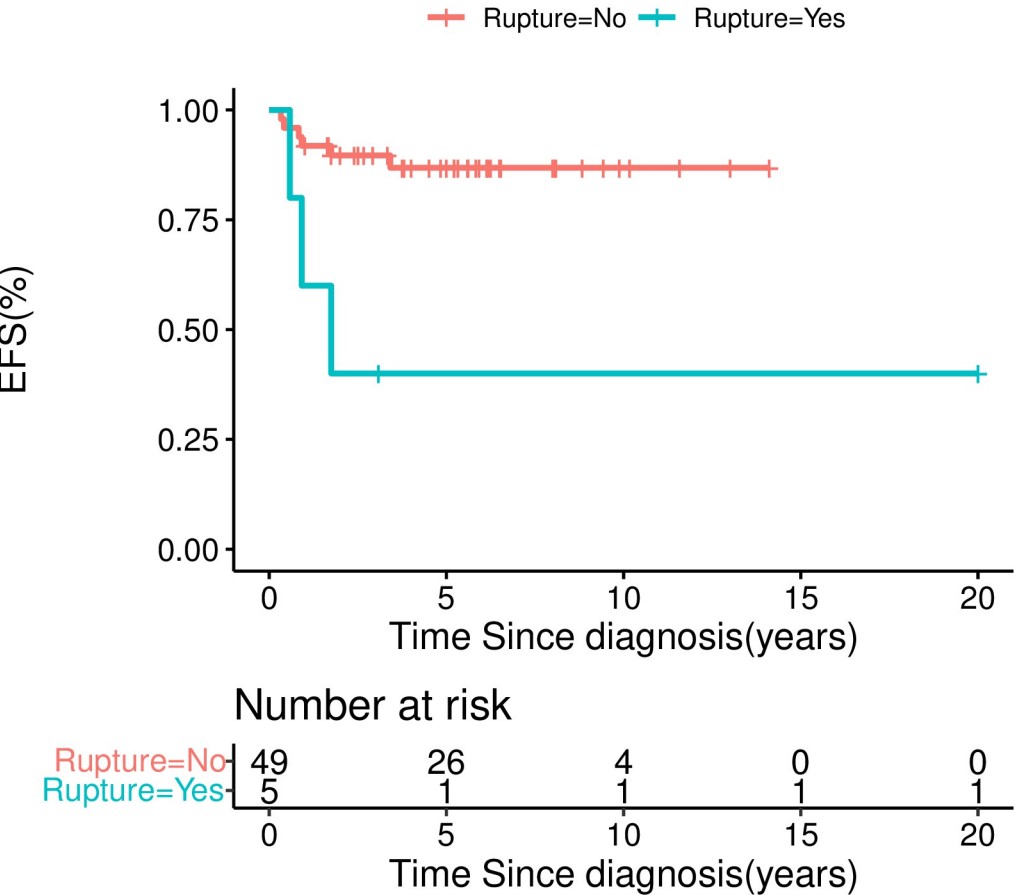

**Fig 6. EFS for CCSK patients by tumor rupture or not.**

detect the risk factors of tumor spill, and investigate the selective use of preoperative chemotherapy or needle biopsy first in such high-risk patients. And the RT doses maybe adjusted for tumor rupture patients. In our study, these patients received 10.8 Gy to the tumor bed. In a United Kingdom Children's Cancer Study Group (UKCCSG) Study, whole abdominal radiotherapy (WART) was administered for patients with disseminated intra-abdominal disease, and those with pre-operative or peri-operative tumor rupture [22]. WART may be more suitable than RT for disseminated abdominal disease and tumor rupture. For stage III patients without tumor rupture, the EFS was about 86% in our study. But the patients with inferior vena cava tumor thrombus also had poor prognosis (41.7%), for such patient, a more intense chemotherapy regimen including carboplatin maybe indicated.

In the NWTS-5 study, patients are administered four-agent chemotherapy called regimen I (cyclophosphamide, etoposide, vincristine, and doxorubicin), for 6 months, along with radiotherapy, regardless of stage [1, 9, 14]. For stage IV, the 5-year EFS was 29% in NWTS-5 [14]. To further improve the prognosis of stage IV CCSK, the AREN0321 study (2006–2013) applied the UHI regimen (CCE/VDC) including carboplatin for patients with stage IV CCSK. In phase II clinical trials, carboplatin has had good effects on relapsed WT and relapsed CCSK [10, 23]. The results of AREN0321 have not been reported. In our study, we followed the NWTS protocol, using carboplatin-containing chemotherapy regimens for stage IV CCSK. The good prognosis for stage IV disease in our study is related to the use of a more aggressive

chemotherapy regimen that included carboplatin, which is not used in NWTS-5, but is used in SIOP. Notably, carboplatin can penetrate the central nervous system, which was also mentioned in the SIOP-2016 protocol [2]. And another impressing result is that in our practice, patients with lung metastasis did not receive lung RT if they achieve compete radiographic response after 6 weeks of systemic treatment (including surgery). None of the patients received lung RT in our study, and only one patient developed a lung relapse (CASE 3 in table) and was salvaged by RT after relapse and are now in second complete remission. This is a contrast to the recommendation in NWTS and SIOP protocol that regardless of response to chemotherapy or surgical treatment, radiotherapy to lung metastasis is indicated in patients with stage IV CCSK [2, 14]. Lung RT may be safely omitted in selected patients who achieve a compete radiographic response after 6 weeks of systemic treatment (including surgery).

The stage I patients in our study exhibited a 100% event-free survival rate, even with two patients who did not receive RT (parents' choice). Similarly, in the NWTS 1–5 protocol, children with revised stage I CCSK had excellent survival rates regardless of whether RT was used, with a 100% survival rate after 17 years of follow-up [24]. Based on these results, the AREN0321 study has plans to avoid radiotherapy to the tumor bed in Stage I CCSK. However, since this tumor carries a high risk of lymph node metastases (20–29% exhibit lymph-node infiltration), eligibility for elimination of radiotherapy should be determined based on the results of lymph-node sampling [10, 14]. Therefore, lymph-node sampling is a rather important part of the surgery.

Recurrences occurred in eight cases in our study. Four of them underwent further treatment, and all four patients achieved a second CR and three are currently alive. These results indicate that patients should not abandon treatment after relapse because long-term survival after recurrence was achievable with intensive multimodal therapy. This has also been confirmed by two other studies. In 2008, Radulescu described eight cases of recurrent CCSK with brain metastasis, and reported that six patients were alive at a median follow-up duration of 30 months from the time of recurrence (range, 24 to 71 months) [25]. The largest study of relapsed CCSK was published in 2014. Among 237 CCSK patients (16%) treated according to SIOP and AIEOP protocols, 37 (16%) developed a relapse. Of these relapsed patients, 59% achieved a second CR, but the five-year event-free survival (EFS) after relapse was 18% [2, 26]. ICE chemotherapy, together with RT and surgery, provides a salvage regimen for recurrent CCSK. It is unclear whether high-dose chemotherapy yields a benefit compared to conventional-dose chemotherapy [25, 26].

The present study had several limitations. It was a retrospective multicenter study, which may suffer from some loss of information, and the lack of long-term follow-ups and genetic analyses of patients.

Recent molecular research confirms that BCOR internal tandem duplication (ITD) is found as a driver mutation in over 80% of cases, and another 5–10% have YWHAE-NUTM2 fusions, which are mutually exclusive [2, 27–30]. Notably, BCOR-ITD tumors constitute an emerging family of aggressive tumors. In addition to CCSK, this family also includes brain tumors, termed central nervous system BCOR-ITD (CNS BCOR-ITD); endometrial tumors, such as high-grade endometrial stromal sarcoma; and bone and soft tissue tumors, such as undifferentiated round cell sarcoma or primitive myxoid mesenchymal tumor of infancy [31, 32]. In infants, BCOR-ITD tumors often present as locally aggressive tumors in the paraspinal region, and are histologically referred to as primitive myxoid mesenchymal tumor of infancy. In toddlers, they often present as CCSK [29]. Interestingly, BCOR-ITD tumors in all locations share a dismal prognosis, with the exception of CCSK. Considering the good prognosis of CCSK, we may consider treating BCOR sarcomas with CCSK-based therapy [29]. Given the current intensity of chemotherapy in CCSK [2], there may be a need for new targeted therapies

in the future. The associated genetic changes may be potential therapeutic targets for future CCSK treatment.

## Conclusion

Tumor rupture was independent poor prognostic factor. Upfront surgery of NWTS strategies can make a definite pathology diagnosis, but how to reduce tumor rupture during surgery and discover risk factors of tumor rupture is important especially in developing countries. The outcomes of patients with stage I–III CCSK in China were comparable to findings in other developed countries.

Better outcomes were achieved in stage IV CCSK by using an intensive chemotherapy regimen including carboplatin in our study, while the result of AREN0321 has not been published. Lung RT may be safely omitted in selected patients who achieve a compete radiographic response after 6 weeks of systemic treatment (including surgery). Treatment should be encouraged even in CCSK cases with metastasis and relapse.

## Supporting information

**S1 Data.**
(XLSX)

## Acknowledgments

The authors are grateful to Dr. Jiaoyang Cai, Hanshan Luan, Yi Le for her help with the data in this paper.

## Author Contributions

**Conceptualization:** Anan Zhang, Xiaojun Yuan, Shayi Jiang, Jing yan Tang.

**Data curation:** Anan Zhang, Xiaojun Yuan, Shayi Jiang, Dongqing Xu, Can Huang, Jing yan Tang, Yijin Gao.

**Formal analysis:** Anan Zhang, Xiaojun Yuan, Shayi Jiang, Can Huang.

**Investigation:** Dongqing Xu.

**Methodology:** Anan Zhang, Xiaojun Yuan, Dongqing Xu, Can Huang, Yijin Gao.

**Project administration:** Anan Zhang, Jing yan Tang, Yijin Gao.

**Resources:** Shayi Jiang, Jing yan Tang.

**Software:** Dongqing Xu.

**Supervision:** Dongqing Xu, Can Huang, Jing yan Tang, Yijin Gao.

**Writing – original draft:** Anan Zhang, Xiaojun Yuan, Shayi Jiang, Dongqing Xu, Can Huang.

**Writing – review & editing:** Anan Zhang, Jing yan Tang, Yijin Gao.

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
