## [Decision Letter · Decision Letter 0]

20 Feb 2024

PONE-D-23-37507Outcomes of children with clear cell sarcoma of kidney following NWTS strategies in Shanghai China (2003–2021)PLOS ONE

Dear Dr. Gao,

Thank you for submitting your manuscript to PLOS ONE. After careful consideration, we feel that it has merit but does not fully meet PLOS ONE’s publication criteria as it currently stands. Therefore, we invite you to submit a revised version of the manuscript that addresses the points raised during the review process.

We look forward to receiving your revised manuscript.

Kind regards,

Gregory Tiao, M.D.

Academic Editor

PLOS ONE

2. PLOS requires an ORCID iD for the corresponding author in Editorial Manager on papers submitted after December 6th, 2016. Please ensure that you have an ORCID iD and that it is validated in Editorial Manager. To do this, go to ‘Update my Information’ (in the upper left-hand corner of the main menu), and click on the Fetch/Validate link next to the ORCID field. This will take you to the ORCID site and allow you to create a new iD or authenticate a pre-existing iD in Editorial Manager. Please see the following video for instructions on linking an ORCID iD to your Editorial Manager account: " ext-link-type="uri" xlink:type="simple">https://www.youtube.com/watch?v=_xcclfuvtxQ".

Additional Editor Comments:

Please address the questions raised in the review

Reviewers' comments:

Reviewer's Responses to Questions

**Comments to the Author**

1. Is the manuscript technically sound, and do the data support the conclusions?

Reviewer #1: Yes

Reviewer #2: Yes

2. Has the statistical analysis been performed appropriately and rigorously? 

Reviewer #1: Yes

Reviewer #2: Yes

3. Have the authors made all data underlying the findings in their manuscript fully available?

Reviewer #1: No

Reviewer #2: Yes

4. Is the manuscript presented in an intelligible fashion and written in standard English?

Reviewer #1: Yes

Reviewer #2: Yes

5. Review Comments to the Author

Reviewer #1: 1. Please describe the sites of metastasis in the 8 patients with Stage IV disease

2. Did LN status impact outcomes in the Stage III patients?

3. There is a mention of only using lung RT in cases where there was residual pulmonary disease after 6 weeks of systemic therapy. Does that mean that those patients with complete radiographic response did not get lung RT? It is important to highlight this point as this is a start contrast to the recommended therapy for lung metastasis in CCSK on NWTS/COG studies

4. Would recommend saying “margin status” rather than “envelope status”

5. Please provide data specifically about intraoperative tumor spill (as opposed to preoperative tumor rupture) as logically these are the cases that may have been possibly avoided with preoperative therapy.

6. Was tumor size correlated with intraoperative tumor rupture?

7. Please discuss how this information may be informative for future clinical practice or clinical trial conduct. It appears that the area for potential improvement may be around those with Stage III disease who may warrant more intensive therapy such as modified UH-1 as is used for Stage IV disease?

8. Please provide numbers at risk for your Kaplan Meier Curves

Reviewer #2: This is a single institutional report on a tumor that has been largely described by cooperative group studies. While the series is large, this does not add anything additional from prior multi-institutional studies, and therapies employed are somewhat outdated. Well-written, but offers no new or insightful information.

6. PLOS authors have the option to publish the peer review history of their article (what does this mean?). If published, this will include your full peer review and any attached files.

Reviewer #1: **Yes: **Nicholas Cost

Reviewer #2: No

---

## [Author Response · Author response to Decision Letter 0]

5 Mar 2024

Response to Reviewers

 Thank you for the constructive advice by editors and reviewers. We agree with the suggestion and have revised according to the comments. We will be happy to edit the paper further based on your helpful comments. We have highlight the changes of the manuscript by red font. 

To Editor:

Thank you for all your help, support and good suggestions!

1)About figure files:

We have resubmitted the figure files according to the guidelines of the journal， the PACE tool, thank you for your help and kindly reminder.

2)About the PLOS ONE’s style requirement:

Thank you for providing us the PLOS ONE style templates of articles and authors and affiliations, we have revised according to the requirements.

3)About the ORCID ID for the corresponding author：

Thank you for reminder and help, we have authenticated the pre-existing ORCID ID in the Editorial Manager.

4)About the Data Availability:

As the journal requested the raw data to replicate all study findings reported in the article, we are willing to upload our excel file that contain the raw data as Supporting Information files. Thank you for your help.

To Reviewer #1:

The reviewer thinks the authors have not made all the data full available.

Response to reviewer: 

Thank you for the reminder. As the journal requested the raw data to replicate all study findings reported in the article, we are willing to upload our excel file that contain the raw data as Supporting Information files. Thank you for your help.

1)Please describe the sites of metastasis in the 8 patients with Stage IV disease

Response to reviewer: 

Thank you for the reminder, we have explained it in the Result Part (Patient characteristics) and also added it in the abstract to facilitate reading.

There were 8 patients with stage IV disease. 6 patients were lung metastasis, one patient was bone metastasis and one was presented with intra-orbital and cervical lymph node metastasis.

2) Did LN status impact outcomes in the Stage III patients?

Response to the reviewer: 

The EFS of LN involvement is 61.5%±13.5%, while EFS of LN negative is 87.5 ±11.7%.

X2=1.561, P=0.212,the fig is in the "response to reviewer"

There was a trend that in stage III patients, LN involvement was related to inferior results, but the P value did not have significance. So we did not put it in the paper. We just mentioned the positive prognostic factor, what is your advice?

3)There is a mention of only using lung RT in cases where there was residual pulmonary disease after 6 weeks of systemic therapy. Does that mean that those patients with complete radiographic response did not get lung RT? It is important to highlight this point as this is a start contrast to the recommended therapy for lung metastasis in CCSK on NWTS/COG studies.

Response to the reviewer:

Yes, it means that if the patients with lung metastasis have a compete radiographic response after 6 weeks of systemic therapy, they did not receive lung RT anymore. We are so thankful for you to highlight this point, it is true that the practice of lung RT is different from COG/SIOP practice, and it brings hope that the prognosis is good without RT.

“None of the six patients with lung metastasis received lung radiotherapy. In one case, the lung metastasis was surgically removed after chemotherapy, and in the other five cases, the lung metastases disappeared within six weeks after chemotherapy.” “And among lung metastasis, only one patient developed a relapse (CASE 3 in table) and are salvaged by RT after relapse and are now in second complete remission.”

We have highlighted this point in the paper (treatment, event) and in the abstract and also added one paragraph to discuss about it, really thank you for pointing out it!!

4)Would recommend saying “margin status” rather than “envelope status”

Response to the reviewer: 

Thank you for your advice, we have revised it.

5）Please provide data specifically about intraoperative tumor spill (as opposed to preoperative tumor rupture) as logically these are the cases that may have been possibly avoided with preoperative therapy.

Response to the reviewer:

Thank you for the advice, we have added it in the paper. Peri-operative tumor rupture occurred in 5 cases, and four cases were intraoperative tumor spill. One patient was seriously ill with preoperative tumor rupture. And we reviewed the operation note that we think some tumor spill was related to the inappropriate surgical practices by surgeons (some were receiving surgery in a not so experienced hospital in China and then transferred to our centers). 

6)Was tumor size correlated with intraoperative tumor rupture?

Response to the reviewer:

This is a great point, thank you. I am sorry not all the operative note provided the exact tumor volume, some just provided the largest diameter of the tumor. From the available data, we can see:

the median tumor size of upfront resection was 10cm (range 9.4cm-14.5cm), 

The median tumor size of tumor spill was 13.1cm(11-18cm)

It seems larger tumor may have higher risk of tumor spill. Therefore, if the surgeons think the tumor has high risk of tumor spill (for example very large tumor volume?), preoperative chemotherapy or needle biopsy maybe more preferred. In the future, we should detect the risk factors of tumor spill, and investigate the selective use of preoperative chemotherapy or needle biopsy first in such high-risk patients. 

What is your suggestion about this point?

Thank you for this question, we have added a little discussion in the paper. 

7）Please discuss how this information may be informative for future clinical practice or clinical trial conduct. It appears that the area for potential improvement maybe around those with stage III disease who may warrant more intensive therapy such as modified UH-1 as is used for stage IV disease?

Response to the reviewer:

A. We think one reason of poor prognosis of stage III in our study is the high proportion of tumor rupture. And tumor spill seems to be a problem in developing country where surgeon is not experienced enough and there are often delays of diagnosis in developing countries. So we think if the surgeons think the tumor has high risk of tumor spill, preoperative chemotherapy or needle biopsy maybe more preferred. In the future, we should detect the risk factors of tumor spill, and investigate the selective use of preoperative chemotherapy or needle biopsy first in such high-risk patients. And the RT doses maybe adjusted for tumor rupture patients.

B. For patients without tumor rupture, the EFS was about 86% in our study. But the patients with inferior vena cava tumor thrombus also had poor prognosis (41.7%), for such patient, a more intense chemotherapy regimen including carboplatin maybe indicated in the future.

Thank you for your good suggestion, we have added a little discussion in the paper. Looking forward to your further advice.

8）Please provide numbers at risk for your Kaplan Meier Curves

Response to the reviewer:

Thank you for your suggestion, we have revised it in all figures.

To Reviewer #2:

This is a single institutional report on a tumor that has been largely described by cooperative group studies. While the series is large, this does not add anything additional from prior multi-institutional studies, and therapies employed are somewhat outdated. Well-written, but offers no new or insightful information.

Response to the reviewer:

Thank you for your suggestions, and let us think about the new insights of our study. 1)First, CCSK is a rare disease, so there are not so many papers involving large series. Our study is not a single institutional report, but a multicenter study with large series in developing country, I think this is one new sight.

2)And secondly, thank you for the reviewers’ wonderful suggestions. In our study, for lung metastasis patient, we are a contrast to the recommendation of NWTS and SIOP practice, we did not give lung RT if the patients achieve CR after 6 weeks of systemic treatment. And the long results are good, for only one of them developed a relapse (1/6). And he was also salvaged by RT after relapse and achieve a second complete remission. So lung RT for lung metastasis maybe omitted in some highly selected patients. 

3)And thirdly, our results of stage 4 were better than the NWTS-5. Because of the rarity, the NWTS has not published their newest results of CCSK.

4)Lastly we find that tumor rupture is a main reason of poor prognosis of stage III patients which should be further pay attention on especially in developing countries. And the RT doses maybe adjusted for tumor rupture patients. We should detect the risk factors of tumor spill in the future, and investigate the selective use of neoadjuvant chemotherapy or needle biopsy first in such high-risk patients.

We think our paper is worth publication, and we are happy if you can give us further suggestions to revise our paper. Thank you very much indeed!!

---

## [Decision Letter · Decision Letter 1]

25 Jun 2024

Outcomes of children with clear cell sarcoma of kidney following NWTS strategies in Shanghai China (2003–2021)

PONE-D-23-37507R1

Dear Dr. Gao,

We’re pleased to inform you that your manuscript has been judged scientifically suitable for publication and will be formally accepted for publication once it meets all outstanding technical requirements.

Kind regards,

Gregory Tiao, M.D.

Academic Editor

PLOS ONE

Additional Editor Comments (optional):

The authors have addressed the concerns raised in previous review

Reviewers' comments:

Reviewer's Responses to Questions

**Comments to the Author**

1. If the authors have adequately addressed your comments raised in a previous round of review and you feel that this manuscript is now acceptable for publication, you may indicate that here to bypass the “Comments to the Author” section, enter your conflict of interest statement in the “Confidential to Editor” section, and submit your "Accept" recommendation.

Reviewer #1: All comments have been addressed

2. Is the manuscript technically sound, and do the data support the conclusions?

Reviewer #1: Yes

3. Has the statistical analysis been performed appropriately and rigorously? 

Reviewer #1: Yes

4. Have the authors made all data underlying the findings in their manuscript fully available?

Reviewer #1: Yes

5. Is the manuscript presented in an intelligible fashion and written in standard English?

Reviewer #1: Yes

6. Review Comments to the Author

Reviewer #1: Thank you for addressing the prior comments. Thank you for providing your experience for the wider medical community.

7. PLOS authors have the option to publish the peer review history of their article (what does this mean?). If published, this will include your full peer review and any attached files.

Reviewer #1: **Yes: **Nicholas G. Cost

---

## [Editor Report · Acceptance letter]

28 Jun 2024

PONE-D-23-37507R1 

PLOS ONE

Dear Dr. Gao, 

I'm pleased to inform you that your manuscript has been deemed suitable for publication in PLOS ONE. Congratulations! Your manuscript is now being handed over to our production team.

Kind regards, 

on behalf of

Dr. Gregory Tiao 

Academic Editor

PLOS ONE